# Viral Nanoparticle System: An Effective Platform for Photodynamic Therapy

**DOI:** 10.3390/ijms22041728

**Published:** 2021-02-09

**Authors:** Shujin Lin, Chun Liu, Xiao Han, Haowei Zhong, Cui Cheng

**Affiliations:** College of Biological Science and Engineering, Fuzhou University, Fuzhou 350108, China; linshujin32@163.com (S.L.); N195720010@fzu.edu.cn (H.Z.)

**Keywords:** photodynamic therapy, viral nanoparticles, photosensitizer

## Abstract

Photodynamic therapy (PDT) is a promising therapy due to its efficiency and accuracy. The photosensitizer is delivered to the target lesion and locally activated. Viral nanoparticles (VNPs) have been explored as delivery vehicles for PDT in recent years because of their favorable properties, including simple manufacture and good safety profile. They have great potential as drug delivery carriers in medicine. Here, we review the development of PDT photosensitizers and discuss applications of VNP-mediated photodynamic therapies and the performance of VNPs in the treatment of tumor cells and antimicrobial therapy. Furthermore, future perspectives are discussed for further developing novel viral nanocarriers or improving existing viral vectors.

## 1. Introduction

Since Niels Finsen utilized red light to activate photosensitizers (PSs) in the skin for prohibiting the production and emission of smallpox pustules in 1960, PSs have been applied for more than 50 years [1,2]. A few years later, Richard Lipson et al. pioneered a new era of photodynamic therapy (PDT) using a compound called “hematoporphyrin derivative” (HpD) as a PS during the treatment of bladder cancer [3,4]. Henceforth, PDT was successfully accepted as a routine treatment for many diseases, with the continuous development and perfection of technology.

The key of photodynamic therapy is the application of a photosensitizer to the patient. The photosensitizer accumulates in the tissue of interest, such as tumor cells, and is activated by light, leading to cell death. In this process, reactive oxygen species (ROS), which are cytotoxic to tumor cells, are efficiently produced from the photochemical reaction, mediated by photosensitizers irradiated by light, and induce cellular inflammatory responses (Figure 1) [5,6]. This pathway involves two types of ROS [7,8,9]: Type I reactions involve electron transfer reactions that produce a series of ROS such as hydrogen peroxide, hydroxyl radicals, and superoxide anions; Type II reactions involve energy transfer, leading to singlet oxygen generation, which is considered the main mechanism [10,11,12]. In the second type, photons are absorbed by the ground state of the photosensitizer. The triplet state of the photosensitizer (^3^PS*), in the presence of molecular oxygen, transfers its excess energy to ground state molecular oxygen (^3^O_2_) via a Dexter-type energy transfer, a process that results in the formation of the PS ground state (^1^PS) and the excited state of molecular oxygen (singlet oxygen, ^1^O_2_) [13,14,15]. Surroundings such as peptides, nucleic acids, and the cell membrane rapidly react with this reactive oxygen, resulting in cell damage and programmed cell death (PCD). Recently, it has been reported that a photoinactivation that is not related to oxygen should be called the “Type III photochemical pathway”. Examples of this type of photodynamic reaction include psoralen and tetracycline, which can achieve antibacterial photodynamic inactivation in the absence of oxygen [16,17].

In order to improve PDT effectiveness, researchers have attempted to investigate novel strategies with advanced phototherapeutic effects and low toxicity. Nanomaterials are a potential carrier of photosensitizers, owing to the advantages such as structural stability, efficient delivery to target cells, high singlet oxygen, and improved photosensitivity. As natural nanoparticles, viral nanoparticles (VNPs) are protein-based materials with higher biocompatibility and tissue-specificity after being decorated with appropriate surface chemistries. Moreover, VNPs are easy to be remolded through genetic engineering and chemical modification. In this review, we review the recent advances of VNP-mediated photodynamic therapies in antimicrobial therapy and the treatment of cancer, and we also provide new perspectives for the development of novel viral nanocarriers for the treatment of tumors.

### 1.1. Application of PDT

PDT has often been used to treat surface disorders such as skin cancer [18]. With the development of nanotechnology drug delivery systems, it can also treat solid malignant cancers such as tumors in lung, stomach, oral, bladder, head, and neck [19,20,21,22,23]. PDT can exhibit a range of mechanisms of antitumor activity, including direct damage to tumor cells or vasculature and stimulation of immune response or inflammation [24,25,26,27]. PDT has shown good effects in the treatment of various diseases, including psoriasis and atherosclerosis [28]. In addition, PDT has been shown to kill microbial cells and treat fungal, bacterial, and viral infections [29,30]. It has been shown to be effective in antiviral therapy, including HIV [31] and herpes [32]. PDT is also widely used to treat acne [33].

There are several advantages in PDT compared to traditional cancer therapy treatment. First, the side effects are weaker than traditional treatments after correct PDT. Surgical procedures are more invasive than PDT, and chemotherapy or radiotherapy results in longer side effects than PDT. In addition, PDT acts on the tumor itself and destroys the tumor-associated vasculature, which greatly impacts tumor death. This allows for repeating treatment several times in one location, with no or little scarring. 

However, there are still some limitations to PDT. First, in disseminated metastases, effective PDT technology is currently unavailable since the irradiated site cannot be the whole body. Highly dense tumors or tumors with necrotic tissue could reduce the efficiency of PDT as tissue oxygenation is critical for treatment effect. Finally, the possibility of light transmission to the target tissue is also complicated. Therefore, due to optical penetration depths of visible light, deep tumors are difficult to treat with PDT.

### 1.2. First- and Second-Generation Photosensitizers

Currently, the Food and Drug Administration (FDA) has approved some PSs for the treatment of certain cancers or precancerous lesions. In general, photosensitizers used in PDT can be divided into first, second, and third generations. 

There are several disadvantages of first-generation photosensitizers that limit their application. For example, it has not been determined how some photosensitizers selectively accumulate in target tissues. Moreover, first-generation photosensitizers, such as HpD and Photofrin, have shown some unfavorable features. Due to their weak light absorption at 630 nm, the optical penetration depths are limited and cannot reach deeper cancer tissues. Its strong phototoxicity to the skin and low metabolic rate in the body means it does not allow long-term treatment.

Second-generation PSs are mainly porphyrin derivatives developed based on porphyrin groups [34]. In addition to being able to stimulate the photosensitizer effectively, PDT also requires the light source to have superior tissue penetration. The light absorption of biological tissues that are located in the wavelength band where hemoglobin absorption and water absorption are both small can provide a so-called treatment window. The penetration depth of green light and blue light is about 2 mm; in contrast, red light (≥600 nm) has better tissue penetration, with a penetration depth of up to 5 mm, and has a better photodynamic therapy effect. Therefore, the light required for photodynamic therapy is generally in the range of 600–800 nm. Porphyrin derivatives can potentially treat a variety of cancer types. The Q-bands of porphyrins are usually near 600 nm and are characterized by low extinction coefficients. In addition, the excitation of singlet oxygen (^1^O_2_) is enhanced to promote tumor cell apoptosis or necrotic effects [35,36,37]. The use of some PSs has been approved by the US FDA and other national regulatory agencies, including benzoporphyrin (Visudyne^®^), hemoporfin (HMME), and texaphyrins [38,39,40].

However, there are still many shortcomings in common second-generation PSs. For example, mTHPC causes long-term photosensitization and accumulation in the skin and has a prolonged biological half-life. 5-Aminolevulinic acid (ALA) and its methylated ester (MAL) have good PDT potential for the treatment of a variety of cancers [41]; however, ALA is limited by its hydrophilic nature, which is not conducive to penetration into deep tissues [42]. It causes adverse reactions in some painful tumors. Studies have shown that the methyl ester of 5-aminolevulinic acid (MAL) is more stable than ALA and penetrates deeply into cells due to its high lipophilicity, which suggests that ALA derivatives with higher lipid solubility than ALA may have greater diffusion capacity [43,44].

Following the study of porphyrins, most studies have turned to a class of photodynamically active natural molecules, such as chlorophyll (CpD) and its derivatives [45,46,47], and some dyes that can be used as potential photosensitizers. Such molecules can be easily obtained from natural resources [48]. Chemical structures of natural photoactive molecules, e.g., chlorin, exhibit excellent photosensitivity compared to the porphyrin derivative, with two additional hydrogens in one pyrrole ring. This results in a wide absorption band and higher tissue permeability. As an important degradation product of chlorophyll-a, purpurin 18 and its derivative (with a polyethylene glycol (PEG) linker) were synthesized as novel photosensitizers (PSs). PEGylated derivatives have higher hydrophilicity, can significantly enhance phototoxicity, and can be used in the PDT of cervical cancer, prostate cancer, pancreatic cancer, and breast cancer [47]. Similarly, the phototoxicity effect of NT–pheophorbide (NT–Pba), a conjugate composed of red luminescent pheophorbide-a and nandrolone (NT), is more pronounced than the original pheophorbide-a [49]. In addition, methylene blue (MB) has low toxicity and can be activated at a wavelength of 666 nm; the yield of singlet oxygen is high [50,51]. These improve the high photodynamic efficiency for the treatment of basal cell carcinoma [52,53]. Another powerful natural photosensitizer is hypericin, which exhibits strong antitumor activity after irradiation, allowing cell death programs to be promoted or offset, thereby increasing the efficiency of PDT [54,55,56,57].

Compared with the first generation, second-generation PSs have a radiation absorption band close to the therapeutic window and deeper permeability; the production of singlet oxygen is also higher. However, the mechanism by which some natural PSs selectively accumulate in tumors is particularly complex, and there are some disadvantages to second-generation PSs. Nanotechnology-based PS delivery systems, such as the third-generation PSs, offer a new approach that overcomes the shortcomings of these PSs and are expected to play a role in the treatment of several diseases.

### 1.3. Third-Generation Photosensitizers

The key factors in the design process of a PDT photosensitizer include solubility and an effective target [58,59]. Compared with traditional small molecular compounds, nanosized carriers have advantages such as long cycle time, high load capacity, and selective targeting [60]. Surface modification of a nanosized carrier or a specific ligand such as a monoclonal antibody or peptide or polyethylene glycol can increase the selectivity and solubility of the photosensitizer, greatly improving the overall efficacy of PDT [61,62].

Recently, nanoparticle-based PS delivery systems have been considered ideal PSs and are widely used in several PDT fields. In these delivery systems, nanoparticles encapsulate or immobilize PSs by covalent or noncovalent interactions [63]. Because of the high surface-area-to-volume ratio, it has the potential for high drug loading [64]. In general, nanoparticles have the following three advantages: (1) they can increase the PS concentration at the target site, thereby reducing the toxicity to other normal tissues; (2) they can improving the solubility in the hydrophobic media of PSs; (3) there is PS delivery at a rate of constant intervals, maintaining a constant and appropriate therapeutic dose at the site of action [63,64,65].

These nano-PSs have two classes: organic and inorganic nanoparticles. Among them, since most viral capsids are in the order of 50–200 nm in diameter [66], they are considered naturally occurring nanoparticles. The virus nanocarrier is a nanoparticle formed by self-assembly of a viral structural protein (envelope protein or capsid protein) and can be used as a basic element and a carrier of a novel nanomaterial. It mimics the external structure and antigenicity of a natural virus with viral structural proteins. Viral nanoparticles play an important role in immunotherapy interventions [67] and imaging applications and serve as vectors for gene delivery and drug delivery [68]. In recent years, viral nanoparticles have also been applied as carriers for photosensitizers. 

## 2. Viral Nanoparticle-Mediated PDT in Tumor Therapy and Bacterial Infection

Since viruses were discovered, they have been known to the public as deadly pathogens. Since the 1950s, researchers have used viruses as a tool to understand the basic life processes of cells and as expression systems in biotechnology [69,70]. More than two decades later, viruses have emerged as vectors for use in gene therapy, cancer treatment, and control of pests in agriculture [71,72,73].

The first nanoviral vector developed has been used in PDT for more than a decade. The first nanoviral vector used in PDT was a plant virus that was used for the treatment of bacteria [74]. Using plant viruses, bacteriophages, and animal viruses, viral vector-mediated PDT has been applied to treat other bacteria and tumor cells such as prostate cancer, breast cancer, and melanoma [75,76,77]. Viral nanoparticles used in PDT can effectively promote the treatment of diseases by PDT.

### 2.1. Phage Nanoparticles

The phages used in PDT as a nanocarrier include filamentous, MS2, and Qβ phages. Filamentous phages are biofibers that specifically infect bacteria [78]. Each phage particle consists of five heritable modified coat proteins enclosing a circular single-stranded (ss) DNA encoding a coat protein. Coat protein (pVIII) acts as the major structure of the phage, with five copies of the two coat proteins (pIII and pVI together, pVII and pIX together) attached to each end [79]. Phage display is a technique to genetically change phage coat protein by fusing the N-terminus of one or more coat proteins to a foreign peptide sequence [80]. This phage, as a novel photosensitizer carrier, has great potential to achieve targeted PDT against cancer cells. Binrui Cao et al. used a pyropheophorbide-a (PPa) chemical conjugation to phage particles specifically for SKBR-3 breast cancer cells to achieve targeted PDT [81]. Shuai Dong et al. observed significant therapeutic effects by conjugating PPA to phage-targeting cell wall mannoprotein MP65 of *Candida albicans*. The nanodrug can significantly increase the accumulation of PPA in *Candida albicans* and trigger a series of apoptotic features such as mitochondrial disorder, S-phase cell arrest, and ROS accumulation [82]. 

As the dodecahedral RNA phage MS2 nanoparticles can transport drugs or biological agents to specific tissues, they are also used in PDT. Phage MS2 nanoparticles can efficiently package drugs into phage capsids to form different types of viral nanoparticles. MS2 protein virus-like particles (VLPs) deliver a wide range of agents with strong immunogenicity and good safety and ensure specific targeting of tissues [83]. MS2 nanocarriers have broad practical application prospects. In 2010, Nicholas Stephanopoulos et al. used phage MS2 to create a multivalent PDT vector to target Jurkat leukemia T-cells [84]. They modified the Jurkat-specific aptamer to the exterior of the capsid, allowing the phage to specifically target the target cell. Up to 180 porphyrins were installed on the interior surface of the self-assembled spherical viral capsid, and the cytotoxicity of the increased singlet oxygen produced upon irradiation killed more than 76% of Jurkat cells. In 2013, Brian A. Cohen and his colleagues loaded up to 250 cationic porphyrins onto the MS2 phage capsid and modified the exterior of the capsid by chemical conjugation with a cancer-targeted nucleic acid aptamer (GTA sequence) [85]. The absorbance calculation of the retentate showed that approximately 60 GTA sequences were loaded on the MS2 capsid with TMAP. In this experiment, MCF-10A cells were used as a control, and life and death staining assays were used to evaluate whether there was cytotoxicity after light induction. The results showed that cell surface receptors on MCF-7 cells could selectively be recognized by GTA but were lower than MCF-10A cells. The results also showed that the porphyrin-MS2 construction could specifically target and kill MCF-7 human breast cancer cells after photoactivation, whereas MCF-10A cells did not die. In addition, cytotoxicity was not observed after the incubation of MCF-7 cells with nontargeted vector viruses loaded with porphyrins. Incubation of MCF-7 cells with an MS2 capsid modified with nontargeting nucleotide sequences did not have any PDT effect, indicating that GTA induces site selectivity in the porphyrin-MS2 construction. Furthermore, the nucleolar aptamers can target other cancers, such as lung, colon, ovarian, and prostate cancers. Moreover, this unique virus-based loading strategy effectively targets the delivery of photosensitizing compounds, providing a viable method for photodynamic cancer treatment of specific sites with biologically derived nanomaterials. 

Viral nanoparticles can be used in a variety of platforms, such as biomedical, material chemistry, and genetic functionalization. Jin-Kyu Rhee et al. simultaneously modified the photosensitizer metalloporphyrin derivative and the glycan ligand that can specifically target the B-cell CD22 receptor cell to the bacteriophage Qβ, so that the modified phage had a targeting function, making the target cell produce a high concentration of localized singlet oxygen [86]. To make icosahedral bacteriophage Qβ VLPs carry photosensitizers and have targeting properties, tetraaryl porphyrin zinc units, which are widely used for the generation of singlet oxygen, were prepared with three amine-terminated hydrophilic arms. The resulting structure was highly soluble in water and not harmful to protein nanoparticles upon attachment. This shows that the function of viral carrier-based photosensitizers is modular, allowing it to be added to other nanoparticles to bind to target cells and produce singlet oxygen. 

These novel phage photosensitizers have opened a new path for PDT, and unique viral vectors provide a solution for effective drug delivery. By using phage display technology, PDT can be utilized to develop different types of phage photosensitizers for various cancers. Both in vivo and in vitro, these specific photosensitizers can effectively eliminate tumors by producing cytotoxic ^1^O_2_ by irradiation with light of the appropriate wavelength at the target site (Figure 2) [76]. However, bacteriophage has limited capacity to take photosensitizers, which could have a big size after modification. There is no in-vivo evidence supporting that bacteriophage-PDT works in diverse cancers. Furthermore, targeting modification on a bacteriophage is difficult due to the genetic capability of this small virus.

### 2.2. Animal Virus Nanoparticles

Nanoparticles constructed by animal viruses play a critical role in the PDT of tumors as a carrier for the delivery of photosensitizing compounds. The high cytotoxicity induced by photosensitizers after illumination is mainly caused by damage to the cell membrane. However, few photosensitizers are localized to the cell membrane; the envelope of the hemagglutinating virus of Japan (HVJ) has a glycoprotein that induces fusion on the host cell membrane, making it a novel drug delivery system that overcomes this limitation. Makoto Sakai et al. found that the addition of HVJ-E enhanced the cytotoxicity of conventional photosensitizers such as 5-aminolevulinic acid (5-ALA) to improve PDT efficacy [87]. To effectively treat drug-resistant prostate cancer, Masaya Yamauchi and coworkers developed an HVJ-E embedded with a PpIX lipid named porphyrus envelope (PE) [75]. PpIX is a photosensitizer widely used in topical PDT. They found that compared with free PpIX lipid or PpIX induced by 5-ALA, PDT using a porphyrin envelope can enhance the uptake of PpIX and the cytotoxicity of PDT. In subsequent research, they demonstrated that rapid delivery of PpIX lipids to target cell membranes can be achieved by using PE [75]. This novel vector selectively and efficiently accumulates in cancer cells, significantly reduces the survival rate of cancer cells, and has potential advantages over traditional carriers, including decreased treatment time, reduced radiation dose, and enhanced tissue penetration.

Surface modification of adenoviral vectors using neutralizing moieties has enabled site-specific gene expression. Junghae Suh et al. used protein phytochrome B (PhyB) and its ligand phytochrome factor (PIF6) to construct the adeno-associated virus (AAV) platform [88]. The rate of viral-based delivery can be controlled by adjusting the ratio of externally applied R and FR light to PhyB and PIF6, respectively. In the treatment of tumors, to systematically use viral therapy and overcome the injection restriction of complex tumors in sensitive organs, Zi-Xian Liao et al. introduced a gene for the production of photosensitive protein KillerRed into a recombinant adeno-associated virus chemically conjugated with iron oxide nanoparticles (approximately 5 nm). Using the adeno-associated virus serotype 2 (AAV2) genome, PDT or light-mediated viral therapy can be achieved [89]. In vivo experiments have shown that this method can significantly inhibit tumor growth. Furthermore, they demonstrated that ironated AAV2 can be magnetically guided, transducing the photosensitive KillerRed protein to achieve PDT irrespective of drug resistance [90].

In addition, the combination of oncolytic vaccinia virus (OVV) and PDT was used to treat primary and metastatic tumors in mice without causing other photodynamic damage to normal tissues of mice [91]. In 2015, Yazan S Khaled et al. combined reovirus oncolytic virus therapy with PpIX-mediated PDT and observed a killing effect on cancer cells [92]. They applied PDT on pancreatic cancer cell lines (PsPC-1 and BXPC-3) and a noncancerous control cell line (HEK293) for 48 h and conjugated the cells with the PpIX prodrug 5-aminolevulinic acid (5-ALA) for 4 h of incubation. The cells were irradiated with visible red-light emitting diodes at 653 nm for 15 min. The trypan blue test and methylthiazole tetrazolium (MTT) were used to analyze the killing effect on cells. Reovirus-using PpIX-mediated PDT resulted in a significant increase in cytotoxicity compared to PDT with reovirus monotherapy, with 100% pancreatic cell death. The results showed that the addition of reovirus before or after PDT showed no significant differences in cytotoxicity. These results provide initial evidence for a new PDT combination therapy. 

Recently, Wenjun Shan et al. loaded a near-infrared fluorescent dye, indocyanine green (ICG), into hepatitis B core protein-like virus particles (HBc VLPs) by regulating the self-assembly process of VLPs, thereby producing Arg-Gly-Asp (RGD)-HBc/ICG VLPs. As one of the few dyes approved for clinical use by the US Food and Drug Administration (FDA), ICG has poor stability in aqueous solution and is easily decomposed by light, which limits its further clinical application [93]. Through the reassembly of the vector, the stability of ICG can be improved, extending its circulation time in the body, increasing the uptake efficiency of the cells, and resulting in efficient delivery to the target site. The results showed that RGD-HBc/ICG VLPs have good biocompatibility and greatly improve cell uptake efficiency. Fluorescence and photoacoustic dual-mode imaging can be achieved in mice, which are expected to be highly sensitive and accurate for tumor detection. Under the illumination of an 808-nm near-infrared laser, RGD-HBc/ICG VLPs can produce photothermal/photodynamic effects, significantly eliminating tumor-loaded tumor tissue (Figure 3). This bioengineered HBc VLP can completely remedy the situation where the application of ICG is restricted due to its insufficient effects… These animal viruses, which are applied to nanocarriers, have greatly advanced the development of viral nanocarriers. 

Animal viruses provide more options in virus–PDT systems, as their biocompatibility is better than bacteriophages. However, animal viruses usually have organophilism, which could limit the function of virus–PDT systems on different kinds of diseases. How to design targeting virus-PDT systems remains to be discovered.

## 3. Plant Virus Nanoparticles

Many plant viruses, such as the cowpea chlorotic mottle virus (CCMV), the tobacco mosaic virus (TMV), and the cowpea mosaic virus (CPMV), are simple in structure, rod-shaped, or have icosahedral symmetry and are relatively easy to design. It is an ideal virus nanocarrier with low production cost and high stability that can be stored at room temperature. In addition to being used to develop vaccines and deliver antitumor drugs, plant viruses have been used to construct vectors for photosensitizers in PDT to treat bacterial infections and improve melanoma treatment [94].

In 2007, Peter A. Suci et al. increased the selectivity of antimicrobial PDT by coupling CCMV with photosensitizers and caused pathogenic *Staphylococcus aureus* inactivation [74]. Functionalized with PSs, a genetically modified CCMV can be used to target *S. aureus* cells. Light-activated killing can be induced by PS functionalized protein under certain light conditions. Natural noninfected nano-Centiana mosaic virus (CPMV) was used as a carrier for PDT to successfully deliver photosensitizers to macrophages and tumor cells, which are specific for immunosuppressive subpopulations of macrophages and target cancer cells. Amy M. Wen et al. found that the conjugation of CPMV/dendritic hybrids can increase the drug-carrying capacity of nanocarriers, effectively eliminating macrophages and tumor cells at low micromolar photosensitizer concentrations, potentially improving melanoma treatment [77]. 

The nucleoprotein component of the tobacco mosaic virus (TMV) is a nanotube with a high aspect ratio that delivers drugs and targets cancer cells. In 2016, antimicrobial photodynamic Zn-EpPor was first applied to cancer and had considerable efficacy compared to that of porphyrin-based PDT [95]. Compared with porphyrin-based PDT, TMV-Zn-EpPor molecules have considerable efficacy and are used in targeted therapies for cancers. In the study, the photosensitizer was successfully loaded into the internal channels of the TMV nanotubes by electrostatic interaction for the treatment of aggressive melanoma. The TMV-Zn-EpPor molecule effectively improved cell uptake and efficacy compared to free photosensitizers. In B16F10 cells, loading Zn-EpPor into TMV improved cell killing efficacy compared to that of free Zn-EpPor alone. The improvement in Zn-EpPor/TMV efficacy was due to increased cellular uptake of Zn-EpPor by TMV delivery. In addition, the pharmaceutical formulation exhibited a good shelf life. Due to the biocompatibility and tumor-homing properties of TMV, photosensitizer–TMV carriers can be used in combination therapy to target melanoma or even other cancers in order for PDT to overcome the limitations of conventional photosensitizers. 

VLPs were formed by many different virus structural proteins in heterologous expression systems such as yeast, *Escherichia coli*, plants, insect cells, and mammalian cells. In 2017, studies found that the VLP of the coat protein of the Physalis mottle virus (PhMV) could be expressed in *E. coli*, with in-vivo activity and low-cost. The photosensitizer Zn-EpPor and the drug crystal violet doxorubicin (DOX) were stably combined with the carrier through noncovalent interaction and were found to be cytotoxic to several cancer cell lines (Figure 4) [96]. The PhMV-derived VLPs are inexpensive to produce and have physical stability. Plant viruses do not replicate in mammals, which indicated plant VLPs are safer than mammalian viruses. However, there are few examples of plant VLPs, and more work is needed.

## 4. Future Outlook

The successful accumulation of photoactive compounds in target tissues is a key factor affecting the therapeutic effects of PDT. The use of viral nanoparticles in PDT has greatly expanded the range of applications of this method. Viral nanoparticles have several advantages: first, they are structurally self-assembling, the protein structure is highly symmetrical and uniform, and they are convenient for production; second, they are biodegradable and biocompatible and have a variety of applications. The most important aspects are that the virus nanoparticles contain no genetic material, are not contagious, and are safe and harmless.

Examples of the treatment of diseases through PDT by using viral nanoparticles as vectors are shown in Table 1. Viral nanoparticles can help accurately transport photosensitizers to target sites to treat a variety of tumors. A novel combination of PDT and gene therapy also has potential in tumor treatment. However, the transfer of photosensitizers or genes is highly dependent on the viral vector used. Therefore, developing new viral nanocarriers or improving existing viral vectors is inevitable in promoting PDT. In the future, additional photosensitizers combined with viral nanoparticles will be explored.

## Figures and Tables

**Figure 1 ijms-22-01728-f001:**
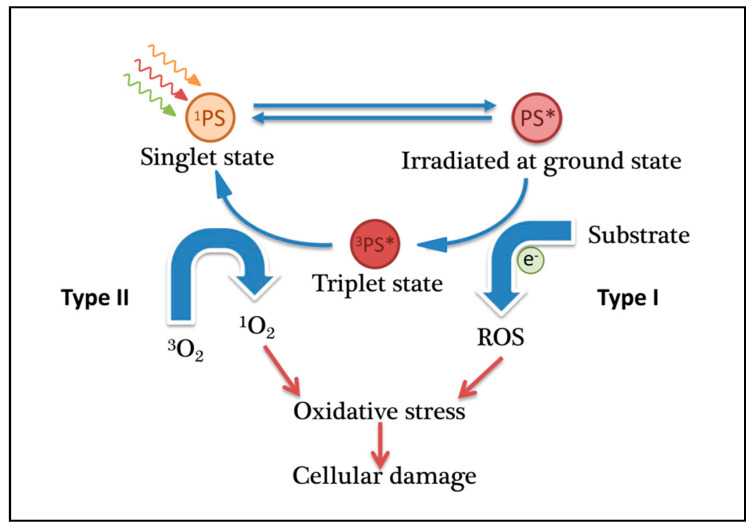
Principle of photodynamic therapy (PDT).

**Figure 2 ijms-22-01728-f002:**
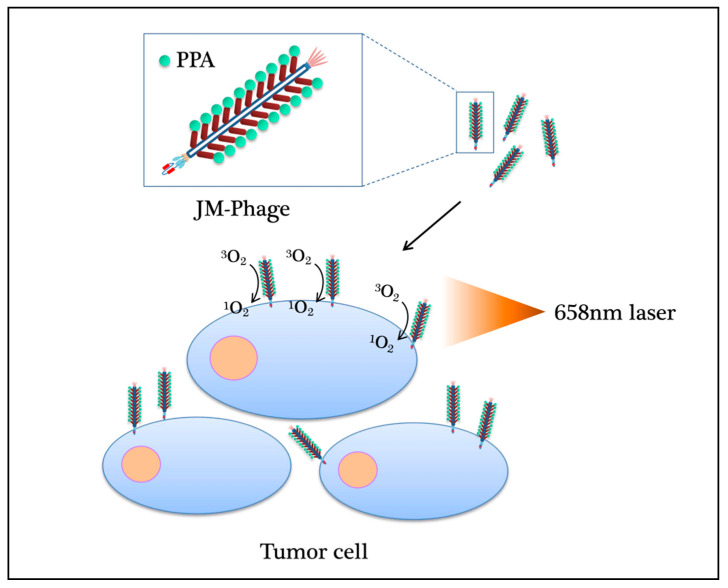
The mechanism of PDT phage photosensitizers for various cancers. PDT can be used to develop different types of phage photosensitizers. Whether in vivo or in vitro, these specific photosensitizers can effectively eliminate tumors by producing cytotoxic ^1^O_2_ by irradiation with light of the 658 nm wavelength at the target site.

**Figure 3 ijms-22-01728-f003:**
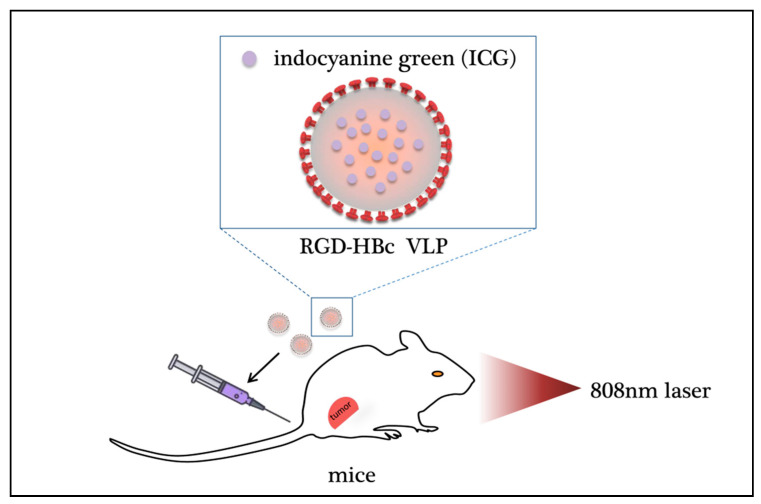
Photodynamic effect of virus nanoparticles carrying antitumor drugs. By regulating the self-assembly process of protein-like virus particles (VLPs), the fluorescent dye ICG can be loaded into a positively charged cavity to prepare RGD-HBc/ICG VLPs, thereby improving the stability of ICG, extending its circulation time in the body and effectively delivering it to the tumor. Under 808 nm near-infrared laser irradiation, RGD-HBc/ICGVLPs can produce photothermal/photodynamic effects and significantly eliminate tumor tissue in mice. ICG: indocyanine green; HBc VLP: hepatitis B core protein virus-like particle; RGD: tripeptide Arg-Gly-Asp.

**Figure 4 ijms-22-01728-f004:**
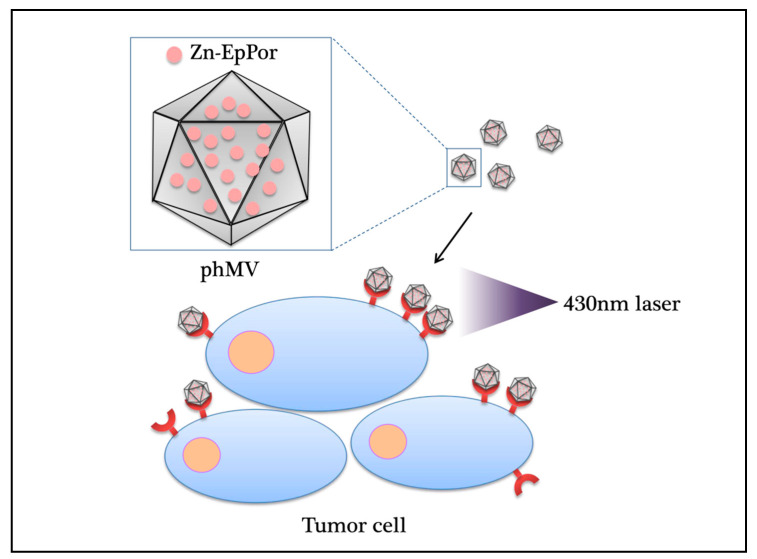
Photodynamic effect of plant virus nanoparticles carrying antitumor drugs. The photosensitizer Zn-EpPor and the drug crystal violet doxorubicin (DOX) stably bind to the VLP of the coat protein of the Physalis mottle virus (PhMV) through noncovalent interactions and are cytotoxic to several cancer cell lines under 430-nm near-infrared laser irradiation. PhMV: Physalis mottle virus; Zn-EpPor: 5-(4-ethynylphenyl)-10,15,20-tris(4-methylpyridin-4-ium-1-yl)porphyrin-zinc(II) triiodide.

**Table 1 ijms-22-01728-t001:** The treatment of diseases through PDT by using viral nanoparticles as vectors.

Types of Virus Nanoparticles	Photosensitizer	Modification	Disease/Cell	PMID
filamentous phage	pyropheophorbide-a (PPa)	SKBR-3 cell-targeting peptide	SKBR-3 breast cancer cells	[81]
JM-phage	pyropheophorbide-a (PPa)	scFv antibody	*Candida albicans*	[82]
phage MS2	porphyrin	Jurkat-specific aptamer	Jurkat leukemia T-cells	[84]
phage MS2	meso-tetra-(4-*N*,*N*,*N*,-trimethylanilinium)-porphine (TMAP)	GTA–G-quadruplex targeting aptamer	MCF-7 human breast cancer cells	[85]
bacteriophage Qβ	metalloporphyrin derivative	BPC derivative (3) of the sialoside Siaα2-6Gal*β*1-4GlcNAc	CHO-CD22+	[86]
HVJ-E	5-aminolaevulinicacid(5-ALA)		the human lung cancer cell line A549 and the murine melanoma cell line B16	[87]
HVJ-E	PpIX lipid	porphyrus envelope(PE)	drug-resistant prostate cancer	[75]
AAV	protein phytochrome B (PhyB)	phytochrome factor (PIF6)	tumor	[88]
AAV2	KillerRed protein	iron oxide nanoparticle	tumor	[89]
Reovirus	5-aminolaevulinicacid(5-ALA)		pancreatic cancer cell line	[92]
vaccinia virus (OVV)	2-[1-hexyloxyethyl-]-2-devinyl pyropheophorbide-a (HPPH)		primary and metastatic tumor	[91]
hepatitis B core protein-like virus(HBc)	indocyanine green (ICG)		tumor	[93]
CCMV	Ru(bpy2)phen-IA	S102C/K42R	*Staphylococcus aureus*	[74]
CPMV	Zn-EpPor	dendron hybrids	macrophages and tumor cells	[77]
TMV	5-(4-ethynylphenyl)-10,15,20-tris(4-methylpyridin-4ium-1-yl)porphyrin-zinc(II) triiodide (Zn-EpPor)	targeting ligands	Melanoma	[95]
PhMV	Zn-EpPor	reactive lysine-N-hydroxysuccinimide ester and cysteine-maleimide chemistries	prostate cancer, ovarian, and breast cancer cell lines	[96]

## Data Availability

Not applicable.

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
