# Peer review of "Viral Nanoparticle System: An Effective Platform for Photodynamic Therapy"

_ijms, 2021, doi:10.3390/ijms22041728_

Round 1
Reviewer 1 Report
The manuscript by Shujin Lin et al. reports a short review on the use of viral particles for the delivery of photosensitizing molecules in PDT applications.
Although the topic is of great interest, this manuscript is too weak as it stands, especially in the first sections.
The whole text needs a thorough revision for the use of the English language, which in many sentences is inaccurate, unclear, or wrong. It is impractical to make a complete list of such language problems, as they are found throughout the whole manuscript. Besides style or language problems, in the first section there are also wrong statements.
While language issues are present, the second and the third sections are outlined better.
I strongly recommend to reconsider this manuscript only after a careful check on the language and a verification of general concepts of PDT.
I will list just a few examples of the problems found in the first pages (this list is by no means reporting all problems in the first pages).
Introduction.
The historical outline appears oversimplified at best. There are a number of reviews the authors could rely on to extract a short historical perspective. Referencing could be improved, please check recent reviews on related subjects.
I list a few here
Gierlich, P., et al. Ligand-Targeted Delivery of Photosensitizers for Cancer Treatment. Molecules 2020, 25 (22), 5317
Wiehe, A., et al. Trends and targets in antiviral phototherapy. Photochemical & Photobiological Sciences 2019, 18 (11), 2565-2612
Callaghan, S.; Senge, M. O. The good, the bad, and the ugly – controlling singlet oxygen through design of photosensitizers and delivery systems for photodynamic therapy. Photochemical & Photobiological Sciences 2018, 17 (11), 1490-1514
Ogilby, P. R. Singlet oxygen: there is indeed something new under the sun. Chem. Soc. Rev. 2010, 39 (8), 3181-3209
Wainwright, M., et al. Photoantimicrobial: are we afraid of the light? The Lancet Infectious Diseases 2017, 17 (2), e49-e55
Planas, O., et al. Newest approaches to singlet oxygen photosensitisation in biological media. In Photochemistry; Albini, A.; Fasani, E., Eds.; The Royal Society of Chemistry: London, 2014.
Hally, C., et al. Photosensitizing proteins for antibacterial photodynamic inactivation. Translational Biophotonics 2020, e201900031
…
Figure 1.
Figure 1 refers to singlet oxygen mediated PDT as Type I, but it is usually termed Type II. Type I is the direct electron transfer process that leads to generation of ROS, and it usually implies a direct reaction with the substrate (not molecular oxygen).
In figure 1 the ground state of the photosensitizer is indicated as PS*, but the * usually refers to the excited state. When the ground state (a singlet state) of the PS absorbs a photon, it is excited to an excited Singlet state PS*. If the authors wish to use the symbol 1PS* for the singlet excited state, emphasizing the spin multiplicity, they may indicate the ground state of the molecules as 1PS.
For a short description of the processes see, e.g., Wiehe, A., et al. Trends and targets in antiviral phototherapy. Photochemical & Photobiological Sciences 2019, 18 (11), 2565-2612
The text in the first page reports the correct terminology, instead.
Line 35
“In the second type, the photons of the ground state light are absorbed by the photosensitizer and converted into the excited triplet state (3O2), which reacts with oxygen to form a highly reactive singlet state (1O2).”
This sentence is full of mistakes. What are “photons of the ground state”? Photons are absorbed by the ground state of the PS.
The excited triplet is that of the PS, not that of molecular oxygen. Molecular oxygen is a triplet state in its ground state.
Moreover, the PS does not react with oxygen. The triplet state of the photosensitizer (3PS*), in the presence of molecular oxygen, transfer its excess energy to the ground state molecular oxygen (3O2) via a Dexter type energy transfer, a process that results in formation of the PS ground state (1PS) and the excited state of molecular oxygen (singlet oxygen, 1O2).
A couple more examples below.
Line 83
“They have higher light absorption and greater tissue penetration in the 650-800 nm range, which speeds up the release of photosensitizers and shortens the period of light shielding to less than two weeks [28].“
The absorption in the red portion has no consequences on the “release” (clearance?) of the photosensitizer
Line 85
“In addition, the oxidation of singlet oxygen (1O2) is enhanced to increase tumor cell apoptosis [28-30]. “
The oxidation is done by singlet oxygen (not to singlet oxygen). The cell death can either be through apoptotic or necrotic effects.
line 86
“The use of some PSs has been approved by the US FDA and other national regulatory agencies, including benzoporphyrin (Visudyne®), hemoporfin (HMME) and Texaphyrins [31-33].”
as it is written, including refers to the national regulatory agencies.
line 97
“The chemical structures of natural photodynamic molecules and dye derivatives exhibit excellent photosensitivity compared to the porphyrin derivative - with two additional hydrogens in one pyrrole ring.”
The molecules are photoactive, not photosensitive (they photosensitize, they are not photosensitized, at least ideally). The two additional hydrogens refer to chlorophyll, but it’s mixed up with dye derivatives.
Author Response
The manuscript by Shujin Lin et al. reports a short review on the use of viral particles for the delivery of photosensitizing molecules in PDT applications.
Although the topic is of great interest, this manuscript is too weak as it stands, especially in the first sections.
The whole text needs a thorough revision for the use of the English language, which in many sentences is inaccurate, unclear, or wrong. It is impractical to make a complete list of such language problems, as they are found throughout the whole manuscript. Besides style or language problems, in the first section there are also wrong statements.
While language issues are present, the second and the third sections are outlined better.
I strongly recommend to reconsider this manuscript only after a careful check on the language and a verification of general concepts of PDT.
I will list just a few examples of the problems found in the first pages (this list is by no means reporting all problems in the first pages).
- Response: Thanks a lot for the reviewer’s comments.
Introduction.
The historical outline appears oversimplified at best. There are a number of reviews the authors could rely on to extract a short historical perspective. Referencing could be improved, please check recent reviews on related subjects.
I list a few here
Gierlich, P., et al. Ligand-Targeted Delivery of Photosensitizers for Cancer Treatment. Molecules 2020, 25 (22), 5317
Wiehe, A., et al. Trends and targets in antiviral phototherapy. Photochemical & Photobiological Sciences 2019, 18 (11), 2565-2612
Callaghan, S.; Senge, M. O. The good, the bad, and the ugly – controlling singlet oxygen through design of photosensitizers and delivery systems for photodynamic therapy. Photochemical & Photobiological Sciences 2018, 17 (11), 1490-1514
Ogilby, P. R. Singlet oxygen: there is indeed something new under the sun. Chem. Soc. Rev. 2010, 39 (8), 3181-3209
Wainwright, M., et al. Photoantimicrobial: are we afraid of the light? The Lancet Infectious Diseases 2017, 17 (2), e49-e55
Planas, O., et al. Newest approaches to singlet oxygen photosensitisation in biological media. In Photochemistry; Albini, A.; Fasani, E., Eds.; The Royal Society of Chemistry: London, 2014.
Hally, C., et al. Photosensitizing proteins for antibacterial photodynamic inactivation. Translational Biophotonics 2020, e201900031
…
- Response: Thank you for your suggestion. We added these references as you asked.
Figure 1.
Figure 1 refers to singlet oxygen mediated PDT as Type I, but it is usually termed Type II. Type I is the direct electron transfer process that leads to generation of ROS, and it usually implies a direct reaction with the substrate (not molecular oxygen).
In figure 1 the ground state of the photosensitizer is indicated as PS*, but the * usually refers to the excited state. When the ground state (a singlet state) of the PS absorbs a photon, it is excited to an excited Singlet state PS*. If the authors wish to use the symbol 1PS* for the singlet excited state, emphasizing the spin multiplicity, they may indicate the ground state of the molecules as 1PS.
For a short description of the processes see, e.g., Wiehe, A., et al. Trends and targets in antiviral phototherapy. Photochemical & Photobiological Sciences 2019, 18 (11), 2565-2612
The text in the first page reports the correct terminology, instead.
- Response: Thank you for your suggestion. We corrected figure 1 as you asked.
Line 35
“In the second type, the photons of the ground state light are absorbed by the photosensitizer and converted into the excited triplet state (3O2), which reacts with oxygen to form a highly reactive singlet state (1O2).”
This sentence is full of mistakes. What are “photons of the ground state”? Photons are absorbed by the ground state of the PS.
The excited triplet is that of the PS, not that of molecular oxygen. Molecular oxygen is a triplet state in its ground state.
Moreover, the PS does not react with oxygen. The triplet state of the photosensitizer (3PS*), in the presence of molecular oxygen, transfer its excess energy to the ground state molecular oxygen (3O2) via a Dexter type energy transfer, a process that results in formation of the PS ground state (1PS) and the excited state of molecular oxygen (singlet oxygen, 1O2).
- Response: Thank you for your suggestion. We corrected this statement as you asked.
A couple more examples below.
Line 83
“They have higher light absorption and greater tissue penetration in the 650-800 nm range, which speeds up the release of photosensitizers and shortens the period of light shielding to less than two weeks [28].“
The absorption in the red portion has no consequences on the “release” (clearance?) of the photosensitizer
- Response: Thank you for your suggestion. We changed the “release” into “effect”.
Line 85
“In addition, the oxidation of singlet oxygen (1O2) is enhanced to increase tumor cell apoptosis [28-30]. “
The oxidation is done by singlet oxygen (not to singlet oxygen). The cell death can either be through apoptotic or necrotic effects.
- Response: Thank you for your suggestion. We corrected them as you asked. The revised part is as follows: ‘In addition, the excitation of singlet oxygen (1O2) is enhanced to increase tumor cell apoptosis or necrotic effects.’
line 86
“The use of some PSs has been approved by the US FDA and other national regulatory agencies, including benzoporphyrin (Visudyne®), hemoporfin (HMME) and Texaphyrins [31-33].”
as it is written, including refers to the national regulatory agencies.
- Response: Thank you for your suggestion. We cannot catch the idea of this question.
line 97
“The chemical structures of natural photodynamic molecules and dye derivatives exhibit excellent photosensitivity compared to the porphyrin derivative - with two additional hydrogens in one pyrrole ring.”
The molecules are photoactive, not photosensitive (they photosensitize, they are not photosensitized, at least ideally). The two additional hydrogens refer to chlorophyll, but it’s mixed up with dye derivatives.
- Response: Thank you for your suggestion. We corrected them as you asked. We replaced “photodynamic” with “photoactive”. The revised part is as follows: ‘As an important degradation product of chlorophyll a, purpurin 18 and its derivative with a polyethylene glycol (PEG) linker were synthesized as novel photosensitizers (PSs). PEGylated derivatives have higher hydrophilicity, can significantly enhance phototoxicity, and be used in PDT of cervical cancer, prostate cancer, pancreatic cancer and breast cancer. Similarly, the phototoxicity effect of NT-pheophorbide a conjugate (NT-Pba) composed of red luminescent pheophorbide a and nandrolone (NT) is more pronounced than the original pheophorbide a. In addition, methylene blue (MB)’.
Reviewer 2 Report
The manuscritp by Lin and Liu et al. entitled Viral Nano-particles system: A efficient platform for photodynamic therapy describes the subject of using virus-derived photosensitizer delivery systems for enhancing the photodynamic therapy. This is an important topic, as the currently clinically approved photosensitizers often lack the properties which would render them selective and thus the implemented therapies often fail because of that. New developed photosensitizers do not always exhibit solubility, selectivity and do not achieve desired concentrations in the tumor sites at a sufficient extent required for efficient therapy. The proposed article fits the scope of the International Journal of Molecular Sciences well.
Unfortunately, the manuscript should be declined publication, as there are too many flaws in it.
My biggest concern is that the articles described in-text are scarce and authors only summarized the results obtained by other authors without any discussion of the results. What is more, there are no conclusions to the manuscript. Without discussion and conclusion to the reviewed articles, the review brings nothing new to the subject.
The viral nanoparticles should be also compared to other currently studied delivery platforms, such as liposomes, metallic nanoparticles, TiO2, ZnO, MOFs and others. Why is their use beneficial? What about the potential immunogenic toxicity associated with proteins?
The introduction section needs serious rewriting by means of the information authors wanted to deliver:
- First, the photodynamic reaction is not caused by photosensitizers (line 30), but is mediated by photosensitizers irradiated by light. The section about the types of photodynamic reactions (lines 31-34) should also be rewritten as it contains some mistakes and is not transparent. Reactions cannot be “composed of substrates” (line 31-32). In line 35 the authors mention “ground state light”, instead of ground state photosensitizer.
- The triplet and singlet state oxygen should be presented as 3O2 and 1O2, and not 3O2 and 1O2, respectively (lines 36, 37, 86).
- The sentence “With the development of nanotechnology drug delivery systems, it can also treat solid malignant cancers such as tumor in lung, stomach, oral, bladder, head and neck [12-16].” should be elaborated as the light delivery seems to be currently the limiting factor for treating solid tumors and delivery vehicles have no effect on this, apart from specific nanoparticles, i.e. afterglow materials or up-conversion nanoparticles.
- What is the dual selectivity you mention in line 64?
- What do you mean by “Highly dense tumors or tumors with necrotic tissue could result in PDT damage” (lines 67-68).
- The sentence “weak absorption at 630 nm and skin sensitivity, which affect its tissue penetration ability and do not allow for long-term treatment.” is unclear and should be rewritten.
- What do you mean by molecular efficiency (line 82)?
- I assume that “they have (…) greater tissue penetration” meant that the absorption range of the photosensitizers offers better light penetration? (line 84) Throughout the article there is no information why the red light region is the therapeutic window for PDT, which is an important information in the context of this review. The same problem can be seen in line 100 “higher tissue permeability” and “enhanced penetration” in line 221.
- How does speeding up the release of photosensitizers (line 84) beneficial for PDT? In line 126 that delivery of PS at a constant rate is an advantage.
- Singlet oxygen cannot be oxidized (line 85).
- You mention ALA, but what about its derivatives for example its methyl esters? (line 92)
- Mentioning dye derivatives seems unclear – you should describe if it considers synthetic dyes. (line 96 and further)
- In lines 106-107 you mention that “Second-generation PSs were developed from natural and known compounds, in contrast to the first generation.” Porphymer sodium and other HpD were developed from natural sources as well by their chemical modification.
- “plays a function in the clinical treatment” (lines 111-112) – actually, there are no clinically approved photosensitizer formulations apart from liposomal ones.
- I disagree with the statement that “most important factors are solubility and effective target in” (lines 114-115). What about the light absorption range? The singlet oxygen generation efficacy? The metabolism and elimination profiles?
- Please explain the “low-molecular-weight reagents” (line 115) as it is not clear.
- Please contextualize that “prevention of degradation” is beneficial (line 117), as the quick degradation after irradiation is beneficial in clinical PDT.
- The paragraphs in lines 128-135 and 137-145 need to be supported by appropriate citations.
- Can reactive oxygen species be accumulated? (line 159)
- Porphyrins cannot be modified on the surface of nanoparticles, did you men conjugated to the surface? (line167)
- The sentence “By efficiently synthesizing each component, Azide-alkyne click chemistry has a single azide-terminated tether” (lines 196-197) is unclear and should be rewritten.
- What do you mean by the “partial nature of the porphyrin”? (line 197)
- If you mention “PpIX lipid” it needs to be described, as currently reading the source publication is needed to understand what does it mean. (line 215)
- What do you mean by “photofired”? (line 273)
- Reference 40 is missing.
- Figures 2-4 are rather vague and should be described more in the caption. Also, they should be mentioned in text where they are described.
Language and style has to be improved throughout the manuscript. Some examples: a efficient (line 2), highly monodisperse (line), drug deliver (line), PSs has (line), high level singlet oxygen (line), including for HIV (line), hematoporphyrin (line), tetoporfin (90), improving the hydrophobicity solubility (126) calculation of the retentate (172), cells were toxic after light induction (174), much lower to (line 176), considered with high solubility (193), bacterial strains should be italic – S. aureus and E. coli (lines 273, 274, 297, 298), are more security (302), N should be in italic in meso-tetra-(4-N,N,N,-trimethylanilinium)-porphine (TMAP) (table 1).
Author Response
The manuscritp by Lin and Liu et al. entitled Viral Nano-particles system: A efficient platform for photodynamic therapy describes the subject of using virus-derived photosensitizer delivery systems for enhancing the photodynamic therapy. This is an important topic, as the currently clinically approved photosensitizers often lack the properties which would render them selective and thus the implemented therapies often fail because of that. New developed photosensitizers do not always exhibit solubility, selectivity and do not achieve desired concentrations in the tumor sites at a sufficient extent required for efficient therapy. The proposed article fits the scope of the International Journal of Molecular Sciences well.
Unfortunately, the manuscript should be declined publication, as there are too many flaws in it.
My biggest concern is that the articles described in-text are scarce and authors only summarized the results obtained by other authors without any discussion of the results. What is more, there are no conclusions to the manuscript. Without discussion and conclusion to the reviewed articles, the review brings nothing new to the subject.
The viral nanoparticles should be also compared to other currently studied delivery platforms, such as liposomes, metallic nanoparticles, TiO2, ZnO, MOFs and others. Why is their use beneficial? What about the potential immunogenic toxicity associated with proteins?
The introduction section needs serious rewriting by means of the information authors wanted to deliver:
First, the photodynamic reaction is not caused by photosensitizers (line 30), but is mediated by photosensitizers irradiated by light. The section about the types of photodynamic reactions (lines 31-34) should also be rewritten as it contains some mistakes and is not transparent. Reactions cannot be “composed of substrates” (line 31-32). In line 35 the authors mention “ground state light”, instead of ground state photosensitizer.
- Response: Thank you for your suggestion. We corrected them as you asked.
The triplet and singlet state oxygen should be presented as 3O2 and 1O2, and not 3O2 and 1O2, respectively (lines 36, 37, 86).
- Response: Thank you for your suggestion. We corrected them following your suggestion.
The sentence “With the development of nanotechnology drug delivery systems, it can also treat solid malignant cancers such as tumor in lung, stomach, oral, bladder, head and neck [12-16].” should be elaborated as the light delivery seems to be currently the limiting factor for treating solid tumors and delivery vehicles have no effect on this, apart from specific nanoparticles, i.e. afterglow materials or up-conversion nanoparticles.
- Response: Thank you for your suggestion. We modified our statements.
What is the dual selectivity you mention in line 64?
- Response: Thank you for your suggestion. We removed this wrong statement.
What do you mean by “Highly dense tumors or tumors with necrotic tissue could result in PDT damage” (lines 67-68).
- Response: Thank you for your suggestion. We modified our statements as ‘Highly dense tumors or tumors with necrotic tissue could reduce the efficiency of PDT as tissue oxygenation is critical for the treatment effect.’
The sentence “weak absorption at 630 nm and skin sensitivity, which affect its tissue penetration ability and do not allow for long-term treatment.” is unclear and should be rewritten.
- Response: Thank you for your suggestion. We corrected them as you asked. ‘Moreover, the first-generation photosensitizers such as HpD and Photofrin have shown some unfavorable features. Due to its weak light absorption at 630 nm, its light penetration is limited and cannot reach deeper cancer tissues.which affect its tissue penetration ability. Its strong phototoxicity to the skin and low metabolic rate in the body make it do not allow for long-term treatment.’
What do you mean by molecular efficiency (line 82)?
- Response: Thank you for your suggestion. We removed this mistake.
I assume that “they have (…) greater tissue penetration” meant that the absorption range of the photosensitizers offers better light penetration? (line 84) Throughout the article there is no information why the red light region is the therapeutic window for PDT, which is an important information in the context of this review. The same problem can be seen in line 100 “higher tissue permeability” and “enhanced penetration” in line 221.
How does speeding up the release of photosensitizers (line 84) beneficial for PDT? In line 126 that delivery of PS at a constant rate is an advantage.
Singlet oxygen cannot be oxidized (line 85).
You mention ALA, but what about its derivatives for example its methyl esters? (line 92)
Mentioning dye derivatives seems unclear – you should describe if it considers synthetic dyes. (line 96 and further)
- Response: Thank you for your suggestion. We corrected them as you asked. “In addition to being able to effectively stimulate the photosensitizer, PDT also requires the light source to have sufficient light wave penetration. The light absorption of biological tissues which located in the wavelength band where hemoglobin absorption and water absorption are both small, can provide a so-called treatment window. Light penetrates longer in the tissue structure. The penetration depth of green light and blue light is about 2mm, while red light (>600 nm) has better tissue penetration, with a penetration degree of up to 5mm, and has a better photodynamic therapy effect. Therefore, the light required for photodynamic therapy is generally in the range of 600-800 nm. Porphyrin derivatives can potentially treat a variety of cancer types and are the most effective sensitizers. They have higher light absorption and stronger ability to penetrate tissues in the 650-800 nm range, which promote the release of photosensitizers and shortens the period of light shielding to less than two weeks [28].”
In lines 106-107 you mention that “Second-generation PSs were developed from natural and known compounds, in contrast to the first generation.” Porphymer sodium and other HpD were developed from natural sources as well by their chemical modification.
“plays a function in the clinical treatment” (lines 111-112) – actually, there are no clinically approved photosensitizer formulations apart from liposomal ones.
I disagree with the statement that “most important factors are solubility and effective target in” (lines 114-115). What about the light absorption range? The singlet oxygen generation efficacy? The metabolism and elimination profiles?
Please explain the “low-molecular-weight reagents” (line 115) as it is not clear.
Please contextualize that “prevention of degradation” is beneficial (line 117), as the quick degradation after irradiation is beneficial in clinical PDT.
- Response: Thank you for your suggestion. We corrected them as you asked.
Can reactive oxygen species be accumulated? (line 159) yes
Porphyrins cannot be modified on the surface of nanoparticles, did you men conjugated to the surface? (line167)
- Response: Thank you for your suggestion. We corrected them as you asked.
The sentence “By efficiently synthesizing each component, Azide-alkyne click chemistry has a single azide-terminated tether” (lines 196-197) is unclear and should be rewritten.
What do you mean by the “partial nature of the porphyrin”? (line 197) If you mention “PpIX lipid” it needs to be described, as currently reading the source publication is needed to understand what does it mean. (line 215) What do you mean by “photofired”? (line 273)
- Response: Thank you for your suggestion. We corrected them as you asked.
Figures 2-4 are rather vague and should be described more in the caption. Also, they should be mentioned in text where they are described.
- Response: Thank you for your suggestion. We corrected them as you asked.
Language and style has to be improved throughout the manuscript. Some examples: a efficient (line 2), highly monodisperse (line), drug deliver (line), PSs has (line), high level singlet oxygen (line), including for HIV (line), hematoporphyrin (line), tetoporfin (90), improving the hydrophobicity solubility (126) calculation of the retentate (172), cells were toxic after light induction (174), much lower to (line 176), considered with high solubility (193), bacterial strains should be italic – S. aureus and E. coli (lines 273, 274, 297, 298), are more security (302), N should be in italic in meso-tetra-(4-N,N,N,-trimethylanilinium)-porphine (TMAP) (table 1).
- Response: Thank you for your suggestion. We corrected them as you asked.
Reviewer 3 Report
This review article brings information/summary on PDT using viral nanoparticles as delivery agents/vehicles for photosensitizer. The review is sound and quite nicely written but it still can be very significantly improved.
Introduction - Type III PDT is missing
Figure 1 - add also type III mechanism and improve the caption, it should be much more extended and the mechanisms should be explained in detail, the figure is a standalone thing and should be understandable without reading the rest of the text - which now it is not
line 61 - "no long-term side effects" - it is not true, what about the skin photosensitivity to daylight, it can last for veeeery long time
line 92 - you write that limitation of using ALA is hydrophilicity, while in line 94 you write that low water solubility of PS in general is limitation, you are contradicting your own statements
line 96 - "dye derivatives" - this is a very broad and empty phrase, give some examples with appropriate references
line 97 - "from natural resources" - give examples as purpurin 18 and its derivatives - this reference must be added:
Molecules 2019 Dec 6;24(24):4477. doi: 10.3390/molecules24244477. PEGylated Purpurin 18 with Improved Solubility: Potent Compounds for Photodynamic Therapy of Cancer
as well as pheophorbide a should be definitely added and also its targeted derivatives, this citation should definitely not be missing: J Mater Chem B 2019 Sep 18;7(36):5465-5477. doi: 10.1039/c9tb01301f. Oxime-based 19-nortestosterone-pheophorbide a conjugate: bimodal controlled release concept for PDT
line 115 - the most important problem is also lack of selectivity
line 157 (and many, many others) - latin words must be in italics (in vitro, in vivo, Candida albicans, etc., etc., there are many words not properly written)
Figure 2 - caption - poor, not saying anything, the caption must be explanatory and understandable stand alone, also the abbreviations from the figure must be explained
explain what KillerRed protein is
line 231 - apoptosis - in which cells? inhibit tumor? in what animal model,e etc., it is not saying much like this
in general, often the type of cells, incubation and irradiation time are missing, improve in whole manuscript
Figure 3 - caption - poor, not saying anything, the caption must be explanatory and understandable stand alone, also the abbreviations from the figure must be explained, structure of ICG should be added
what photofired means?
Staph. a. - in latin
line 296 - empty general sentence - you should elaborate more on this
line 301 - several cell lines - empty not saying anything, explain
as for a review, the article has only very little references
Figure 4 - caption - poor, not saying anything, the caption must be explanatory and understandable stand alone, also the abbreviations from the figure must be explained, value and a unit - there must be a space in between
Table 1 - you should either extend the table and provide the PS structures (where possible) or do some extra figure, in which you would show their chemical structure, it is more than desired, they must be added
instead of PMID - add references in normal citation format
plus "modification" - add the peptide sequences (either in supplementary information or in the main article, but they should be added)
Author Response
This review article brings information/summary on PDT using viral nanoparticles as delivery agents/vehicles for photosensitizer. The review is sound and quite nicely written but it still can be very significantly improved.
- Response: Thanks a lot for the reviewer’s comments.
Introduction - Type III PDT is missing
Figure 1 - improve the caption, it should be much more extended and the mechanisms should be explained in detail, the figure is a standalone thing and should be understandable without reading the rest of the text - which now it is not
- Response: Thank you for your suggestion. We corrected them as you asked. The added introduction of Type III PDT as follows: ‘Recently, it has been reported that a photo-inactivation that is not related to oxygen should be called the "type III photochemical pathway". Examples of this type of photodynamic reactions include psoralen and tetracycline that can achieve antibacterial photodynamic inactivation in the absence of oxygen.’
line 61 - "no long-term side effects" - it is not true, what about the skin photosensitivity to daylight, it can last for veeeery long time
- Response: Thank you for your suggestion. We replaced “no long-term side effects” with “the side effects are weaker than traditional treatments”.
line 92 - you write that limitation of using ALA is hydrophilicity, while in line 94 you write that low water solubility of PS in general is limitation, you are contradicting your own statements
- Response: Thank you for your suggestion. We corrected them as you asked.
line 96 - "dye derivatives" - this is a very broad and empty phrase, give some examples with appropriate references. line 97 - "from natural resources" - give examples as purpurin 18 and its derivatives - this reference must be added:
Molecules 2019 Dec 6;24(24):4477. doi: 10.3390/molecules24244477. PEGylated Purpurin 18 with Improved Solubility: Potent Compounds for Photodynamic Therapy of Cancer
- Response: Thank you for your suggestion. We corrected them as you asked.
line 115 - the most important problem is also lack of selectivity. line 157 (and many, many others) - latin words must be in italics (in vitro, in vivo, Candida albicans, etc., etc., there are many words not properly written)
- Response: Thank you for your suggestion. We corrected them as you asked.
Figure 2 - caption - poor, not saying anything, the caption must be explanatory and understandable stand alone, also the abbreviations from the figure must be explained
- Response: Thank you for your suggestion. The revised part is as follows:
“Figure 2. The mechanism of PDT phage photosensitizers for various cancers. PDT can be used to develop different types of phage photosensitizers. Whether in vivo or in vitro, these specific photosensitizers can effectively eliminate tumors by by producing cytotoxic 1O2 by irradiation with light of the 658 nm wavelength at the target site.”
explain what KillerRed protein is
- Response: Thank you for your suggestion. KillerRed is a gene for production the photosensitive protein.
line 231 - apoptosis - in which cells? inhibit tumor? in what animal model,e etc., it is not saying much like this
- Response: Thank you for your suggestion. We corrected them as you asked.
Figure 3 - caption - poor, not saying anything, the caption must be explanatory and understandable stand alone, also the abbreviations from the figure must be explained, structure of ICG should be added
- Response: Thank you for your suggestion. The revised part is as follows:
Figure 3. Photodynamic effect of virus nanoparticles carrying anti-tumor drugs By regulating the self-assembly process of VLPs, the fluorescent dye ICG can be loaded into a positively charged cavity to prepare RGD-HBc/ICG VLPs, thereby improving the stability of ICG, extending its circulation time in the body and effectively delivering it to the tumor. Under 808 nm near-infrared laser irradiation, RGD-HBc/ICGVLPs can produce photothermal/photodynamic effects and significantly eliminate tumor tissue in mice. ICG: indocyanine green; HBc VLP: hepatitis B core protein virus-like particle; RGD: tripeptide Arg-Gly-Asp.
what photofired means?
Staph. a. - in latin
line 296 - empty general sentence - you should elaborate more on this. line 301 - several cell lines - empty not saying anything, explain. Figure 4 - caption - poor, not saying anything, the caption must be explanatory and understandable stand alone, also the abbreviations from the figure must be explained, value and a unit - there must be a space in between
- Response: Thank you for your suggestion. We corrected them as you asked.
Table 1 - you should either extend the table and provide the PS structures (where possible) or do some extra figure, in which you would show their chemical structure, it is more than desired, they must be added
- Response: Thank you for your suggestion. We describe the PS structures in text.
plus "modification" - add the peptide sequences (either in supplementary information or in the main article, but they should be added)
- Response: Thank you for your suggestion. Table 1 already take full of page, the peptide sequences can be download from references. It’s meaningless to write here. Also these are kinds of modification not only peptide.
Round 2
Reviewer 1 Report
In the first lines, the historical perspective has not been touched, I suggest revising it along the lines suggested my my first report
line 24. HpD is hematoporphyrin derivative
line 36, sentence not clear: ”type II reaction…in its ground state”
lines 37-42 please check superscripts. Please check also the sentence, which appears to be cut at some point.
Figure 1, the sequence of events in the scheme is not correct, please revise it, use as a guide Figure 2 in ref 15.
In general, the paper still presents language issues that have not been fixed in the revision.
Also added text should be checked for language issues (see, e.g. line 96)
Author Response
In the first lines, the historical perspective has not been touched, I suggest revising it along the lines suggested my my first report.
- Response: Thank you for your suggestion. In general, we summarized the limitations on PDT,made a future outlook arried out with PDT.’
In Phage nanoparticles part, we discussed as following,
‘These novel phage photosensitizers open a new path for PDT, and unique viral vectors provide a solution for effective drug delivery. By using phage display technology, PDT can be utilized to develop different types of phage photosensitizers for various cancers. Both in vivo and in vitro, these specific photosensitizers can effectively eliminate tumors by producing cytotoxic 1O2 by irradiation with light of the appropriate wavelength at the target site (figure.2)[76]. However, bacteriophage has limited capacity to take more kind of photosensitizers which could has more biological function. There is no in vivo evidence supported that bacteriophage-PDT could work in diverse cancers. Furthermore, targeting modification on bacteriophage is difficult due to the genetic capability of this small virus.’
In animal virus nanoparticles part, we discussed as following,
‘These animal viruses, which are applied to nanocarriers, have greatly advanced the development of viral nanocarriers. Animal virus provide more option in virus-PDT system, as its biocompatibility is better than bacteriophage. However, animal virus usually has organophilism which could limited the function of virus-PDT system on different kinds of disease. How to design targeting virus-PDT system is remained to be discovered.’
In plant virus part, we discussed as following,
‘The PhMV-derived VLPs are inexpensive to produce and have physical stability. Plant viruses does not replicate in mammals, which indicated plant VLPs are safer than mammalian virus. However, there are few examples of plant VLPs, and more work is needed.’
line 24. HpD is hematoporphyrin derivative
- Response: Thank you for your suggestion. We replaced “hemoporphyrin derivative” with “hematoporphyrin derivative”.
line 36, sentence not clear: ”type II reaction…in its ground state”
- Response: Thank you for your suggestion. We made more concise sentence ‘In the second type, photons are absorbed by the ground state of photosensitizer.’
lines 37-42 please check superscripts. Please check also the sentence, which appears to be cut at some point.
- Response: Thank you for your suggestion. We checked and corrected them with the superscript.
Figure 1, the sequence of events in the scheme is not correct, please revise it, use as a guide Figure 2 in ref 15.
- Response: Thank you for your suggestion. We corrected Figure 1 as you asked.
In general, the paper still presents language issues that have not been fixed in the revision. Also added text should be checked for language issues (see, e.g. line 96)
- Thank you for your suggestion. We checked and revised the language issues of the whole article.We corrected line96 as following: ‘They can absorb more light which has stronger ability to penetrate tissues in the 650-800 nm range, and promote the effect when the period of light shielding is less than two weeks’.
Reviewer 2 Report
In the revised version of the article, some of my earlier comments were addressed, but not all of them. The article still presents a little scientific value to the reader due to the lack of discussion of the results.
Certain fragment of the text, although corrected do contain new mistakes and errors, like 1O2 without the superscript; methylated ALA is mentioned just to address the comment but the description afterwards is unchanged and refers to ALA itself. Many sentences mentioned in the previous review were corrected but some of them still need rewriting as the pointed out mistakes were exchanged by others. PpIX lipid is explained as protoporphyrin lipid, which cannot be counted as improvement at all.
I appreciate the expanding of the figures’ captions but I do not see any difference in the changed figures.
All in all, I recommend the manuscript to be rejected as it does not meet the high requirements for the articles published in International Journal of Molecular Sciences.
Author Response
in the revised version of the article, some of my earlier comments were addressed, but not all of them. The article still presents a little scientific value to the reader due to the lack of discussion of the results.
Certain fragment of the text, although corrected do contain new mistakes and errors, like 1O2 without the superscript; methylated ALA is mentioned just to address the comment but the description afterwards is unchanged and refers to ALA itself. Many sentences mentioned in the previous review were corrected but some of them still need rewriting as the pointed out mistakes were exchanged by others. PpIX lipid is explained as protoporphyrin lipid, which cannot be counted as improvement at all.
- Response: Thank you for your suggestion. We corrected 1O2 with the superscript. We have supplemented the comparison between MLA and ALA between lines 109-111 in the last revised version of the article. According to ref 75, PpIX lipid is protoporphyrin IX lipid .
I appreciate the expanding of the figures’ captions but I do not see any difference in the changed figures.
- Response: Thank you for your suggestion. We changed figures with higher resolution.
All in all, I recommend the manuscript to be rejected as it does not meet the high requirements for the articles published in International Journal of Molecular Sciences.
Reviewer 3 Report
The authors have fulfilled all that was required and answered all raised questions.
Author Response
The authors have fulfilled all that was required and answered all raised questions.
- Response: Thanks a lot for the reviewer’s comments.
Round 3
Reviewer 2 Report
The manuscript has improved after the revision by authors, however there are some aspects that still need to be addressed:
- The photosensitizer does not have to be delivered systematically (line 10).
- Rather than hydroxyl groups, you should use the term hydroxyl radicals (line 34).
- You mention higher biocompatibility (line 53) – were there any studies on the subject? Introduction of a foreign peptide into the body might result in immune reaction. Please support this statement with a reference.
- The use of the phrase “in the clinical treatment” (line 63) would suggest that all the uses and properties mentioned afterwards until the end of the paragraph are clinically approved. Please correct it. The same goes for the term applied (line 171).
- You mention that the porphyrin derivatives are the “most effective sensitizers” (lines 99-100). Please back up this statement with a reference. The Q-bands of porphyrins are usually near 600 nm and are characterized by low extinction coefficients.
- “Chemical structures of natural photoactive molecules exhibit excellent photosensitivity compared to the porphyrin derivative - with two additional hydrogens in one pyrrole ring.” (lines 118-119) – please use the term “chlorin”.
- The paragraph in lines 132-135: Please rewrite it. The absorption band in the visible region does not have to be wider – it has to be in the therapeutic window. You mention all the three generations of the photosensitizers but never actually define the criteria for each generation.
- “Surface modification with a nanosized carrier” (lines 142-143) – did you mean “Surface modification of a nanosized carrier”?
- What do you mean by high yields? (line 149)
- “improving the hydrophobic solubility of PSs” (lines 151-152) – please rewrite, as it currently suggests that the effect will be better solubility in hydrophobic media.
- “PS delivery at a constant rate, hold on a constant therapeutic dose” (lines 152-153) – please rewrite, as it is unclear. Why is the constant rate important? The irradiation is not continuous over prolonged period of time.
- “simultaneously sensitized the photosensitizer metalloporphyrin derivative” (line 216) – please rewrite. The photosensitizer cannot be sensitized.
- “However, bacteriophage has limited capacity to take more kind of photosensitizers which could have more biological function” (lines 229-231) – please rewrite, as it is unclear.
- Protoporphyrin IX lipid is a chemically modified PpIX in which the peripheral vinyl groups were hydrolyzed and appended with long aliphatic chains. It isn’t a very popular photosensitizer by itself, so I think that is why it should be explained in the text. Alternatively, simply describing it as PpIX derivative would suffice.
- “5-ALA-induced freePpIX lipid” (line 252) – please rewrite, as it is unclear.
- What does the “(5)” stand for? (line 370).
- English and style: for (line 9); treatment (28); is (31); penetration (78); carried (79); its (87); “light penetration is limited and cannot reach deeper cancer tissues .which affect its tissue penetration ability” (88-89); make it does (90); “the light source to have sufficient light wave penetration” (93); Light penetrates longer in the tissue structure (95); degree (97); promote the effect (101); defects (107); target in (141); concentration of (150); constructtion (line 205); “prepare” (220) – hydrophilic arms cannot be used to prepare porphyrins, unlike to attach them to the nanosystem; considered have (221-222); benign (224); enhanced penetration (256) – did you mean enhanced internalizacion?; conjugated (274); reovirus-using not reovirus using (277); The experiment also evaluated after PDT (278-279); RGD-HBc/ICG VLPs instead of RGD. -HBc/ICG VLPs (284); limited application of the drug ICG effect (294); the paragraph in lines 296-299; inactivated by photodynamic (318).
Author Response
The manuscript has improved after the revision by authors, however there are some aspects that still need to be addressed: The photosensitizer does not have to be delivered systematically (line 10). Rather than hydroxyl groups, you should use the term hydroxyl radicals (line 34). Response: Thank you for your suggestion. We deleted “systematically” in line 10 and replaced “hydroxyl groups” with “hydroxyl radicals”. You mention higher biocompatibility (line 53) – were there any studies on the subject? Introduction of a foreign peptide into the body might result in immune reaction. Please support this statement with a reference. Response: Thank you for your suggestion. We checked and corrected them in the revised article. The biocompatibility we are talking about here means that nanoparticles can be decorated with appropriate surface chemistries to enhance their biocompatibility, tissue-specificity, and therefore overall efficacy. The use of the phrase “in the clinical treatment” (line 63) would suggest that all the uses and properties mentioned afterwards until the end of the paragraph are clinically approved. Please correct it. The same goes for the term applied (line 171). Response: Thank you for your suggestion. The revised part in line 63 is as follows: ‘PDT has shown good effects in the treatment of various diseases including psoriasis, and atherosclerosis’. We deleted the word "clinical" in line 171. You mention that the porphyrin derivatives are the “most effective sensitizers” (lines 99-100). Please back up this statement with a reference. The Q-bands of porphyrins are usually near 600 nm and are characterized by low extinction coefficients. Response: Thank you for your suggestion. We removed the "most effective sensitizers" in the revised article. We corrected them as you asked. “Chemical structures of natural photoactive molecules exhibit excellent photosensitivity compared to the porphyrin derivative - with two additional hydrogens in one pyrrole ring.” (lines 118-119) – please use the term “chlorin”. Response: Thank you for your suggestion. We used the term chlorin. The paragraph in lines 132-135: Please rewrite it. The absorption band in the visible region does not have to be wider – it has to be in the therapeutic window. You mention all the three generations of the photosensitizers but never actually define the criteria for each generation. Response: Thank you for your suggestion. The revised part is as follows: ‘Compared with the first generation, the second-generation PSs has the radiation absorption band close to the therapeutic window.’ Instead of defining the criteria for each generation, we listed the characteristics of each generation of the photosensitizers. “Surface modification with a nanosized carrier” (lines 142-143) – did you mean “Surface modification of a nanosized carrier”? Response: Thank you for your suggestion. We replaced “with” with “of”. What do you mean by high yields? (line 149) “improving the hydrophobic solubility of PSs” (lines 151-152) – please rewrite, as it currently suggests that the effect will be better solubility in hydrophobic media. Response: Thank you for your suggestion. We removed the "high yields" in the revised article. We checked and corrected them in the revised article. The revised part is as follows: “(2)they can improving the solubility in hydrophobic media of PSs”. “PS delivery at a constant rate, hold on a constant therapeutic dose” (lines 152-153) – please rewrite, as it is unclear. Why is the constant rate important? The irradiation is not continuous over prolonged period of time. Response: Thank you for your suggestion. Here we want to show that the drug delivery dose can be controlled within an appropriate range by adjusting the irradiation time. The revised part is as follows: “(3) PS delivery at a rate with constant interval , maintaining a constant and appropriate therapeutic dose at the site of action” “simultaneously sensitized the photosensitizer metalloporphyrin derivative” (line 216) – please rewrite. The photosensitizer cannot be sensitized. Response: Thank you for your suggestion. We replaced “sensitized” with “modified”. “However, bacteriophage has limited capacity to take more kind of photosensitizers which could have more biological function” (lines 229-231) – please rewrite, as it is unclear. Response: Thank you for your suggestion. We rewrite it as following, However, bacteriophage has limited capacity to take photosensitizers which could has big size after modification. Protoporphyrin IX lipid is a chemically modified PpIX in which the peripheral vinyl groups were hydrolyzed and appended with long aliphatic chains. It isn’t a very popular photosensitizer by itself, so I think that is why it should be explained in the text. Alternatively, simply describing it as PpIX derivative would suffice. Response: Thank you for your suggestion. We removed the "Protoporphyrin IX lipid" in the revised article. “5-ALA-induced freePpIX lipid” (line 252) – please rewrite, as it is unclear. Response: Thank you for your suggestion. The revised part is as follows:“They found that compared with the free PpIX lipid or PpIX induced by 5-ALA, PDT using a porphyrin envelope can enhanced uptake of PpIX and cytotoxicity of PDT.” What does the “(5)” stand for? (line 370). Response: Thank you for your suggestion. We removed the "(5)" in the revised article. English and style: for (line 9); treatment (28); is (31); penetration (78); carried (79); its (87); “light penetration is limited and cannot reach deeper cancer tissues .which affect its tissue penetration ability” (88-89); make it does (90); “the light source to have sufficient light wave penetration” (93); Light penetrates longer in the tissue structure (95); degree (97); promote the effect (101); defects (107); target in (141); concentration of (150); constructtion (line 205); “prepare” (220) – hydrophilic arms cannot be used to prepare porphyrins, unlike to attach them to the nanosystem; considered have (221-222); benign (224); enhanced penetration (256) – did you mean enhanced internalizacion?; conjugated (274); reovirus-using not reovirus using (277); The experiment also evaluated after PDT (278-279); RGD-HBc/ICG VLPs instead of RGD. -HBc/ICG VLPs (284); limited application of the drug ICG effect (294); the paragraph in lines 296-299; inactivated by photodynamic (318). Response: Thank you for your suggestion. We checked and corrected them in the revised article.